# Uncovering microbiomes of the rice phyllosphere using long-read metagenomic sequencing
Sachiko Masuda[1], Pamela Gan[1], Yuya Kiguchi[2,3,4], Mizue Anda [2], Kazuhiro Sasaki[5,6], Arisa Shibata[1], Wataru Iwasaki [2], Wataru Suda[4] & Ken Shirasu [1,7] ✉

The plant microbiome is crucial for plant growth, yet many important questions remain, such as the identification of specific bacterial species in plants, their genetic content, and location of these genes on chromosomes or plasmids. To gain insights into the genetic makeup of the rice-phyllosphere, we perform a metagenomic analysis using long-read sequences. Here, 1.8 Gb reads are assembled into 26,067 contigs including 142 circular sequences. Within these contigs, 669 complete 16S rRNA genes are clustered into 166 bacterial species, 121 of which show low identity (<97%) to defined sequences, suggesting novel species. The circular contigs contain novel chromosomes and a megaplasmid, and most of the smaller circular contigs are defined as novel plasmids or bacteriophages. One circular contig represents the complete chromosome of a difficult-to-culture bacterium *Candidatus Saccharibacteria*. Our findings demonstrate the efficacy of long-read-based metagenomics for profiling microbial communities and discovering novel sequences in plant-microbiome studies.

The phyllosphere, encompassing plant surfaces and internal structures such as leaves, flowers and fruits allow diverse microbes to form complex communities collectively known as the plant microbiome[1,2]. The phyllosphere microbiome plays a crucial role in host growth and health[3]. Beneficial microbes within the phyllosphere can directly promote plant growth through improved nutrient uptake, pathogen suppression, and disease resistance[3]. The composition and function of the phyllosphere microbiome are influenced by biotic elements such as host plant genetics[4], plant defense mechanisms[5] and microbe-microbe interactions[6], as well as abiotic factors like temperature[7] and UV radiation[8], and human activities like agricultural practices[9]. While individual members of the plant microbiome can exhibit beneficial characteristics, the overall impact of the microbiome on plant health is complex and cannot be predicted from individual microbial taxa.

To gain deeper insights into the ecological and biological features of phyllosphere microbiomes, both culture-dependent and culture-independent methods, particularly next-generation sequencing technologies, have increasingly been employed as a tool to analyze the genetic makeup of complex microbial communities[1]. Those approaches provide a catalog of the microbial diversity and functional potential within a given

community[1]. For example, phyla such as Proteobacteria, Bacteroidetes, Firmicutes, and Actinomycetes are prevalent in *Arabidopsis thaliana* plants[10]. Notably, Proteobacteria often constitute up to half of the community composition, with *Methylobacterium*, plant growth-promoting bacteria, being a dominant genus in the plant phyllosphere[11]. However, those methods have their limitations and biases. Cultivation methods offer a limited view since many microorganisms cannot be cultured in vitro[12]. Traditional short-read (<500 bp) gene amplicon sequencing, typically of the variable regions of 16S rRNA genes or other single genes poses serious limitations for accurate identification[13], i.e., the PCR-biased amplification efficiency such as primer bias. Although short-read metagenomics can reconstruct nearly complete plasmids and chromosomes[14], reconstructing complex genomes remains challenging. Therefore, a large proportion of short reads that cannot be mapped to a reference genome results in the loss of potentially useful information[15]. In contrast, long-read metagenomics has the potential to generate longer contigs, thus improving genome reconstruction, taxonomic assignment, and revealing previously undiscovered sequences, including circular genomes and extrachromosomal elements. For instance, long-read metagenomic sequencing of the human gut

[1]RIKEN Center for Sustainable Resource Science, Kanagawa, Japan. [2]Department of Integrated Biosciences, Graduate School of Frontier Sciences, The University of Tokyo, Tokyo, Japan. [3]Cooperative Major in Advanced Health Science, Graduate School of Advanced Science and Engineering, Waseda University, Tokyo, Japan. [4]RIKEN Center for Integrative Medical Sciences, Kanagawa, Japan. [5]Institute for Sustainable Agro-ecosystem Services, Graduate School of Agricultural and Life Sciences, The University of Tokyo, Tokyo, Japan. [6]Japan International Research Center for Agricultural Sciences, Ibaraki, Japan. [7]Graduate School of Science, The University of Tokyo, Tokyo, Japan. ✉e-mail: ken.shirasu@riken.jp

microbiome has uncovered a higher number of plasmids than previously reported[13]. We anticipate that long-read metagenomics will be a valuable approach for exploring plant microbiome metagenomes.

Plasmids and bacteriophages have a role in shaping microbial communities by providing novel capabilities[16,17], such as the well-studied VirB/VirD4 Type 4 Secretion System (T4SS) found in both pathogenic and symbiotic bactera in plant-microbiome interactions[18–21]. The VirB/VirD4 system, known for its presence on the Ti plasmid in *Agrobacterium tumefaciens*, enables pathogenic bacteria to invade host plants by transporting effector proteins and manipulating the host's immune system. On the other hand, the VirB/VirD4 system in root-nodulating bacteria is involved in protein translocation and can have a host-dependent effect on symbiosis[22]. Cultivating bacteria from natural environments in labs is challenging, leading to reliance on metagenome assembly. However, this process is limited: the depiction of each element (chromosome, plasmid, and bacteriophage) hinges on its representation in the DNA mix and the occurrence of shared repeats among various elements[23,24]. Consequently, resulting genomes assembled from elements scarce in the environment tend to be incomplete, requiring high coverage for accurate assembly[25]. While bioinformatics tools aid in reconstructing plasmid sequences from short reads, achieving a contiguous assembly remains difficult. Despite their ecological importance, our understanding of plasmid and bacteriophage sequences from phyllosphere microbiome is still limited.

In this study, we used long-read metagenomics to better understand the genetic makeup of the rice (*Oryza sativa*) microbiome. We enriched microbes from the phyllosphere and established a genomic DNA extraction method for long-read sequencing. Then, using reads from the Pacbio Sequel II sequencer, we reconstructed 26,067 contigs, including novel circularized chromosomes, plasmids, and bacteriophages. Notably, we identified the complete chromosome of the candidate phylum *Candidatus Saccharibacteria*. Our results demonstrate that long-read-based metagenomics provides a powerful tool for profiling plant-associated microbial communities.

## Materials and methods
### Sampling and bacterial cell enrichment
Rice plants (*Oryza sativa* cultivar 'Koshihikari') were grown in an experimental paddy field at the Institute for Sustainable Agro-ecosystem Services, Graduate School of Agricultural and Life Sciences, The University of Tokyo (35°74′N, 139°54′E). Plants were sampled before heading on 6 August 2018 (8-week-old plants), and their roots and aerial parts were separated and stored at −80 °C. The aerial parts were ground using a Roche GM200 grinder (2000 rpm 15 sec Hit mode, 8000 rpm 30 sec Cut mode, and 8000 rpm 15 sec Cut mode) with dry ice. The ground aerial parts were homogenized in the bacterial cell extraction buffer containing 50 mM Tris-HCl (pH 7.5), 1% Triton X-100 and 2 mM 2-mercaptoethanol in a blender (7010HS, Waring Laboratory) on high speed with 1 min for three times. The homogenate was filtered with Miracloth (Merck), and the filtrate was centrifuged at 500 rpm. The supernatant was transferred to a new tube, centrifuged, and the pellet was washed with bacterial cell extraction buffer. The washed pellet was resuspended in the 50 mM Tris-HCl (pH 7.5), and then the suspension was overlaid on Nycodenz (Axis-Shield Diagnostics) solution (8 g Nycodenz in 10 ml of 50 mM Tris-HCl (pH 7.5)) and centrifuged at 9000 rpm for 40 min at 10 °C. After centrifugation, the interface was collected as a bacterial cell fraction. An equal volume of sterilized water was mixed to the collected fraction, and the bacterial fraction was obtained by centrifugation[26]. DNase was added to the bacterial cell fraction to digest the plant-derived genomic DNA, and 0.5 M EDTA was added to stop the DNase reaction. Then, DNase-treated bacterial cells were stored at −80 °C before use.

### DNA extraction
Genomic DNA was extracted from the enriched plant-associated microbes using enzymatic lysis. The cells were lysed by the addition of 20 mg/ml Lysozyme (Sigma-Aldrich), 10 μl of Lysostaphin (>3000 unit/ml, Sigma-

Aldrich), and 10 μl of Mutanolysin (>4000 units/ml, Sigma-Aldrich), and incubated for 3 h at 37 °C. SDS (20%, Sigma-Aldrich) and Proteinase K (10 mg/ml, Sigma-Aldrich) were then added, and DNA was purified with cetrimonium bromide and phenol-chloroform. DNA was then incubated with RNase for 30 min at 37 °C (Nippongene) and dissolved in TE buffer at 4 °C. The sequencing library was constructed and sequenced within a week of DNA extraction. We also extracted genomic DNA using a Fast DNA spin kit (MP-Biomedicals) for mechanical lysis from the enriched microbes. DNA fragmentation was assessed using pulsed-field gel electrophoresis.

### Sequencing of the V4 regions of 16 S rRNA genes
The V4 regions of 16S rRNA genes were amplified using the primers 515 F (5′-ACA CTC TTT CCC TAC ACG ACG CTC TTC CGA TCT GTG CCA GCM GCC GCG GTA A-3′) and 806 R (5′-GTG ACT GGA GTT CAG ACG TGT GCT CTT CCG ATC TGG ACT ACH VGG GTW TCT AAT-3′) and sequenced with an Illumina MiSeq v3 platform. The first 20 bases of primer sequences were trimmed from all paired reads. For low-quality sequences, bases after 240 bp and 160 bp of forward and reverse primer sequences, respectively, were truncated, the processed reads were aligned to the SILVA123 dataset[27], and their taxonomy was provisionally determined using Qiime2 v 2018.11.0[28].

### Sequencing, assembly, and gene annotation of the plant microbiome
SMRTbell libraries for sequencing were constructed according to the manufacturer's protocol (Part Number 101-693-800 Version 01) without shearing. The libraries were cut off at 20 kbp using the BluePippin size selection system (Sage Sciences). Libraries were sequenced on two SMRT Cells 8 M (Pacific Biosciences). We removed contaminating plant sequences, subreads showing more than 80% identity and 80% length coverage according to minimap2 v 2.14[29] to 'Nipponbare' as the reference rice genome, and PacBio's internal control sequences from subreads. 'Nipponbare' was used as the reference genome[30] because the draft genome of 'Koshihikari' is highly fragmented, with an average read length of 468 bp[31]. The remaining subreads were assembled using Canu v 1.8[32] with minOverlapLength=2200, minReadLength=2200, genomeSize=100 m, corOutCoverage=10000, corMhapSensitivity=high, corMinCoverage=0[33]. After assembly, we removed contaminated contigs derived from internal controls and reference genome using the same method, and artificial contigs with long stretches of G, C, A or T by calculation of GC contents with seqkit v 0.11.0[34]. For quality assessment of the contigs[13], we aligned the error-corrected reads generated during assembly to the contigs with pbmm2 v 1.2.1 (Pacific Biosciences) with maximum best alignment 1 and minimum concordance percentage 90 set as parameters, and extracted the contigs with a depth >5. Contig circularity was determined as the sequences at the start and end of the contigs were overlapped[33]. For confirmation of quality, error-corrected reads were aligned to contigs using pbmm2 with the same parameters as for contig quality assessment, then gaps were assessed using IGV browser v 2.8.2[35]. Quast v 5.0.2[36] was used to evaluate the quality of genome assemblies. Functional annotation of bacterial genes was conducted using PROKKA v 1.14.6 in the metagenomic mode[37], COG database (BLASTP with the *e* value lower than 1e-05), Interproscan v 5.46–81.0[38] (with the *e* value lower than 1e-05) and kofamscan v 1.3.0[39]. Augustus v 3.4.0 was used to annotate the genes of fungal genomes[40].

### Estimation of microbial composition using 16S rRNA genes
We obtained bacterial 16S rRNA gene sequences from NCBI BioProject 33175 (Bacteria) and 33317 (Archaea), removed those ≤1400 bp in length, and clustered the remaining sequences using CD-HIT v 4.8.1 (≥97% identity)[41]. The resulting curated 16S rRNA gene database contains 11,782 distinct sequences. In the long read-based assembly data, 16S rRNA genes longer than 1400 bp on contigs having an average read depth ≥5 were aligned to our curated 16S rRNA gene database to obtain the maximum number of target hits. Alignments with <95% length coverage were removed. We used 16S rRNA genes with ≥97% identity as the top hits for

approximating bacterial community composition at the species level. We counted the depth of the contigs carrying 16S rRNA genes to estimate their abundance.

Full-length 16S rRNA genes were amplified using the primers 27F (5′-AGR GTT YGA TYM TGG CTC AG -3′) and 1492 R (5′- RGY TAC CTT GTT ACG ACT T -3′), and sequenced on a SMRT cell 1 M v3. Circular consensus sequences (>3 paths) were constructed and demultiplexed using SMRTLink v 8.0.0 with default parameters. Primers and chimeric reads were removed from demultiplexed CCS reads using dada2 v 1.16[42], and reads ≥1400 bp were extracted. Full-length 16S rRNA amplicons were aligned with our curated database to assign taxa and to estimate bacterial community composition at the species level (≥97% identity). 16S rRNA gene sequences in the metagenomic data were aligned with MAFFT v 7.475[43] using default parameters. A phylogenetic tree was constructed using RAxML v 8.2.12[44] and visualized using FigTree v 1.4.4 (http://tree.bio.ed.ac.uk/software/figtree/).

### Classification of circular contigs
Circular chromosomes or megaplasmids contigs were classified according to the criteria of diCenzo and Finan[45]. Comparisons of circular contigs, the sequences to reference genomes obtained from NCBI database, Plsdb v 2020_06_29[46] and pVOGs[47] were plotted with mummerplot. Contig completeness and contamination of chromosome were calculated using checkM v 1.2.2[48]. Interproscan was used to classify those contigs as chromosome, plasmid or bacteriophage for searching genes encoding replication proteins, DnaA, RepA, and virus-related genes. Kofamscan was used to annotate VirB/VirD4 systems on plasmids. Plasflow[49], similarity search using NCBI database and Mob-typer v 3.0.0 via MOB-suite[50] were also used for plasmid prediction and host identification. Virsorter2 v 2.2.2 (a score cut off >0.8)[51] predicted bacteriophage origins of contigs, while CheckV v 0.7.0[52] assessed the quality of viral sequence contigs. Taxonomy assignment at the genus level was reliable only when a contig's genes aligned with a different species within a genus, exceeding one-fourth of the contig's genes and originating from a single phylum. In other cases, taxonomy remained unassigned. However, Mob-typer sometimes provided taxonomy assignments where the nt database did not. For these contigs, taxonomy was tentatively assigned based on Mob-typer results. Gene maps were created with 'ggplot2' in R.

### Assignment of metagenomic assembled genomes and large size contigs
The metagenomic assembled genomes (MAGs), except for the circular contigs, were reconstructed as outlined by two tools, MetaBat v 2:2.15[53] with -m 5000 -x 5 and MaxBin v 2.2.7[54] with -min_contig_length 5000. Then, those bins were refined using the "bin_refinement" of MetaWRAP v 1.2.1[55] with -c 20 -x 10[56]. To assign taxonomy of these MAGs, average nucleotide identity (ANI) was used to assign the contigs to taxa with GTDB-tk v 1.1.1 using default parameters[57]. Additionally, we also assigned taxonomy of large size contigs (>1 Mbp) that were not included in the initial binning process, following the same methodology used for MAGs. To provisionally assign taxa to contigs that were not assignable using either of the above methods, the annotated genes on the contigs were aligned to the nt database in NCBI using BLASTN with ≥80% identity and ≥80% length coverage. Any contigs with uncertain taxonomy were not classified, according to the criteria in our "Classification of circular contigs' section."

### Taxonomic assignment and predicted gene function
We aligned all predicted genes to the COG database with an $e$ value <1e-05 to predict gene function[13]. A similarity search of the annotated genes was conducted against the nt database in NCBI using BLASTN with ≥95% identity and ≥90% coverage. From these genes, we extracted those which were identified by species, and counted both the number of genes and contigs that carried these genes.

### Genomic features of Candidatus Saccharibacteria
We obtained the 16S rRNA genes of *Candidatus Saccharibacteria* from the NCBI database, removed sequences ≤1400 bp, and clustered the remaining using CD-HIT at 97% identity. The genomic sequences of RAAC3 (GenBank: CP006915.1), *aalborgensis* (S_aal, GenBank: CP005957.1), GWC2 (GenBank: CP011211.1), YM_S32 (GenBank: CP025011.1), and *Candidatus Nanosynbacter lyticus* strain TM7x (GenBank: CP007496.1) were obtained for genomic comparisons. Kofamscan and BlastKOALA[58] were used to predict metabolic features. Average amino acid identity was calculated using the Kostas lab AAI calculator with default parameters (http://enve-omics.ce.gatech.edu/aai/).

### Statistics and reproducibility
The rice-phyllosphere samples were randomly taken at the same day from the same compartment in an experimental paddy field. Rice plants were mixed since we obtain the required amount of gDNA for PacBio Sequel II sequencing.

### Reporting summary
Further information on research design is available in the Nature Portfolio Reporting Summary linked to this article.

## Results
### DNA preparation for long-read metagenomics from rice phyllosphere
Comparing the genomic extraction methods such as enzymatic and mechanical cell lysis using pulsed-field gel electrophoresis showed that enzymatic lysis yielded intact chromosomes, whereas mechanically lysed cells yielded fragmented DNA (Supplementary Fig. 1). Comparisons of bacterial community composition indicated that the two methods gave similar results (Supplementary Fig. 2), although some phyla were more highly represented in one method than in the other, possibly due to differences associated with cell lysis of different taxa. Given the importance of intact genomic DNA for long-read sequencing, we chose to use enzymatic lysis for further analyses.

### Long-read metagenomic sequencing of leaf-associated microbes
We sequenced genomic DNA from the leaf-associated microbes using two Sequel II 8 M cells, yielding 140 Gbp of data with a mean read length of 17 kbp and a mean library size of 15 kbp (Supplementary Data), indicating that our DNA extraction method was suitable for PacBio long-read sequencing. We obtained 26,067 contigs in total after assembly, with an $N_{50}$ of 128 kbp, including 142 circular contigs (Table 1). A previous study reported that PacBio contigs with ≥1 and 5 read depths had ≥98.5% and 99.4% identity, respectively, when aligned to short-read contigs. We thus were able to define 13,050 contigs with a depth of more than 5 as high quality contigs. These contigs represented approximately half of the total, and represented about 80% of the total reads (Table 1). Additionally, more than 90% of the total nucleotides were found in the set of contigs with a length of ≥50 kbp. Importantly, all large size contigs (≥1 Mbp) were of high quality (Table 1).

### Table 1 | Summary of assembly results

| Assembly results | Total contigs | High quality contigs |
|---|---|---|
| Number of the contigs | 26,067 | 13,050 |
| Total nucleotides (bp) | 1,763,254,923 | 1,521,823,352 |
| Number of nucleotides (bp, ≥50 kbp) | 1,390,656,635 | 1,370,599,048 |
| Largest contig size (bp) | 8,528,088 | 8,528,088 |
| Contig $N_{50}$ | 127,510 | 185,687 |
| Predicted CDS | 2,046,382 | 1,674,802 |
| Number of contigs ≥1 Mbp | 172 | 172 |
| Number of circular contigs | 142 | 132 |

These data suggest that nucleotide sequences obtained from the rice phyllosphere microbiome are reliable for estimating bacterial community composition and their functions within the community.

## Estimation of microbial composition using 16S rRNA genes in long-read metagenomics

We extracted 16S rRNA gene sequences ≥1400 bp in length and ≥95% coverage of the top hits from high quality contigs from the metagenomic data. A total of 669 16S rRNA genes were identified on 561 contigs, representing 4.4% of high quality contigs (Supplementary Data). A phylogenetic tree was used to summarize the taxonomy of the detected 16S rRNA genes in the metagenome (Fig. 1). Many of 16S rRNA genes were clustered with top-hit sequences at various taxonomic ranks, such as the species and genus levels, but some were independently clustered. The 669 16S rRNA genes clustered into 166 bacterial species, 121 of which had ≤97% identity (widely used for bacterial species definition[10]) with any organism in the database, suggesting that they represent novel species (Supplementary Data). Clustering the 16S rRNA genes using a threshold for bacterial taxonomy (97%) showed that 463 of the 16S rRNA genes on 369 contigs were ≥97% identical to sequences attributable to known taxa, but 206 were ≤97% identical to known taxa (Table 2). Among the latter, 16 were ≤82 and, 1 was <78%, suggesting that they potentially represent a novel order and class, respectively (Table 2).

Taxonomic analysis of the high-confidence identity 16S rRNA sequences (463 sequences with ≥97% identity) identified 59 bacterial species (Supplementary Data). For example, *Curtobacterium pusillum*, plant-growth-promoting bacteria, which is known was the most abundant species; 67 of 16S rRNA genes were identified on 51 contigs with a relative abundance of 11.2%. Similarly, 8 species of *Methylobacterium* were identified in the high-confidence identity sequences, with five species (*M. aquaticum*, *M. indicum*, *M. radiotolerans*, *M. komagae* and *M. persicinum*) having more than 10 16S rRNA genes (Supplementary Data). The number of contigs containing 16S rRNA genes was mostly lower than the number of 16S rRNA genes, indicating that these bacteria carry multiple 16S rRNA gene copies

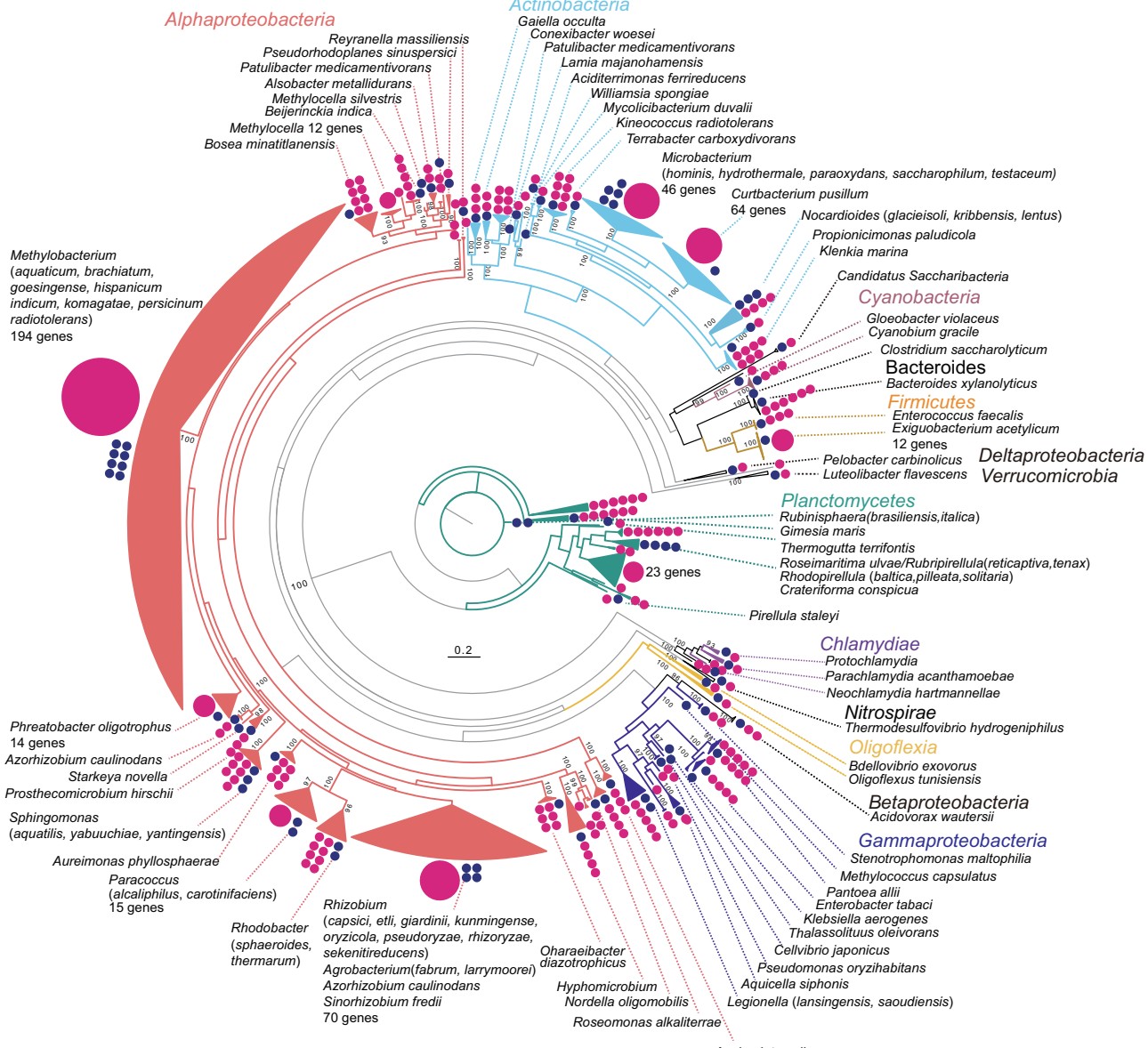

**Fig. 1 | Overview of the phylogeny of 16S rRNA genes detected in the metagenome.** Pink circles indicate the number of 16S rRNA genes detected in the metagenome for each major branch: large pink circles with numbers represent branches with more than ten genes, small pink circles each represent one gene. Blue circles represent the number of top species/genus identities of metagenome-derived 16S rRNA genes.

(Supplementary Data). In particular, nine of the 16S rRNA genes from *Exiguobacterium acetylicum* were detected on a single 1.7 Mbp contig (Contig ID: RRA86345 in Supplementary Data).

We compared the taxonomic profile of bacterial community composition based on the long-read metagenome, full-length 16S rRNA gene amplicon sequences, and the short-read 16S rRNA sequencing. Among the 6678 reads obtained by full-length 16S rRNA gene amplicon sequencing, 4800 reads (67%) have ≥97% identity to taxonomically classified 16S rRNAs, a result similar to the metagenomic data (Supplementary Fig. 3). We identified 112 taxa to the species level, with *Ensifer mexicanus* was the most abundant. There were 19 species with relative abundance >1%, 15 of which were also detected in the long-read metagenome. The combined relative abundance of the 19 species was 57.8% in the metagenome and 59.1% in the 16S rRNA nearly full-length amplicon data sets (Supplementary Data).

The relative abundance of *Actinobacteria* was about twice as high in the long-read metagenome (28.6%) than in either the nearly full-length 16S rRNA amplicon dataset (15.8%) or the V4 region dataset (15.2%). Comparing the relative abundance of *Actinobacteria* showed that the relative abundance of *Micrococcales* in the metagenome was about twice that of PCR-based amplicon sequencing (Fig. 2). The difference may be because certain classes of *Actinobacteria* were difficult to detect using universal primers, even though the target sequences are identical[59]. Previous studies also showed that the V4 region is less reliable for classifying *Actinobacteria* sequences[60]. Therefore, our results suggest that long-read metagenome

## Table 2 | 16S rRNA genes above the threshold for bacterial taxonomy detected in the metagenome

| Identity (%) | Taxonomic rank | number of 16S rRNA genes | number of contigs |
|---|---|---|---|
| 97 | Species | 463 | 369 |
| 94.5 | Genus | 61 | 58 |
| 86.5 | Family | 127 | 121 |
| 82 | Order | 16 | 16 |
| 78 | Class | 1 | 1 |

sequencing provide more accurate identification about dominant bacterial communities in the aerial parts of rice, particularly for *Actinobacteria*.

## Taxonomic assignment of predicted genes
We identified a total of 2,046,382 predicted genes in the metagenome (Table 1 and Supplementary Fig. 4). Of these putative genes, 364,262 had an e-value of 1.0e-05 or less and were annotated using the COG database. Approximately 20% of the genes were categorized as poorly characterized group 'R' (11.8%, general function prediction only) or 'S' (9.6%, function unknown). 8% of the genes were annotated as amino acid metabolism (E), 6.5% as carbohydrate transport and metabolism (G), and 6.2% as energy production and conversion (C). Among these five categories, 50−70% of the genes were derived from *Alphaproteobacteria*, particularly *Methylobacterium* (12.9−17.3%). The putative genes from Planctomycetes were the second most abundant (9.6−18.0%, Supplementary Fig. 4). These results showed that *Methylobacterium* is the dominant genus in the rice phyllosphere, supporting the bacterial community composition predicted by 16S rRNA genes (Fig. 1).

## Classification and taxonomic assignment of circular contigs
We reconstructed 142 circular contigs ranging in size from 8.5 kbp to 4.3 Mbp, with the GC content from 36.8% to 75.2% (Fig. 3 and Supplementary Data). Among them, six contigs were larger than 1 Mbp, with five and one being assigned as bacterial chromosomes and a megaplasmid, respectively (Fig. 3). The taxonomy of these genomes can be tentatively assumed by 16S rRNA gene sequence similarity and/or ANI, though the nucleotide sequences of some contigs do not match those of sequenced strains (Supplementary Fig. 5). Five contigs (RRA2267, RRA3045, RRA6539, RRA85519, and RRA944769) carry 16S rRNA genes with 87.4–99% identity to the top hit (Fig. 3). Notably, the complete chromosome sequence of *Rhizobium giardini* (RRA85519) was obtained for the first time, which can serve as a valuable reference for this species. We classified only one contig as a megaplasmid (RRA6539; Fig. 3), as it showed high similarity to a plasmid (NZ_CP049244.1) of *Rhizobium pseudoryzae*, which also carries 16S rRNA genes on both its chromosome and plasmid. The other three contigs (RRA2267, RRA3045, and RRA85519) were placed at the genus rank by ANI, all of which were consistent with the 16S rRNA-based assessment

**Fig. 2 | Relative abundance of 16S rRNA genes detected in the metagenome, compared to 16S rRNA gene full-length amplicon sequences and short reads. A** Relative abundance of bacterial phyla and (**B**) relative abundance of the class *Actinobacteria* for each sequencing method.

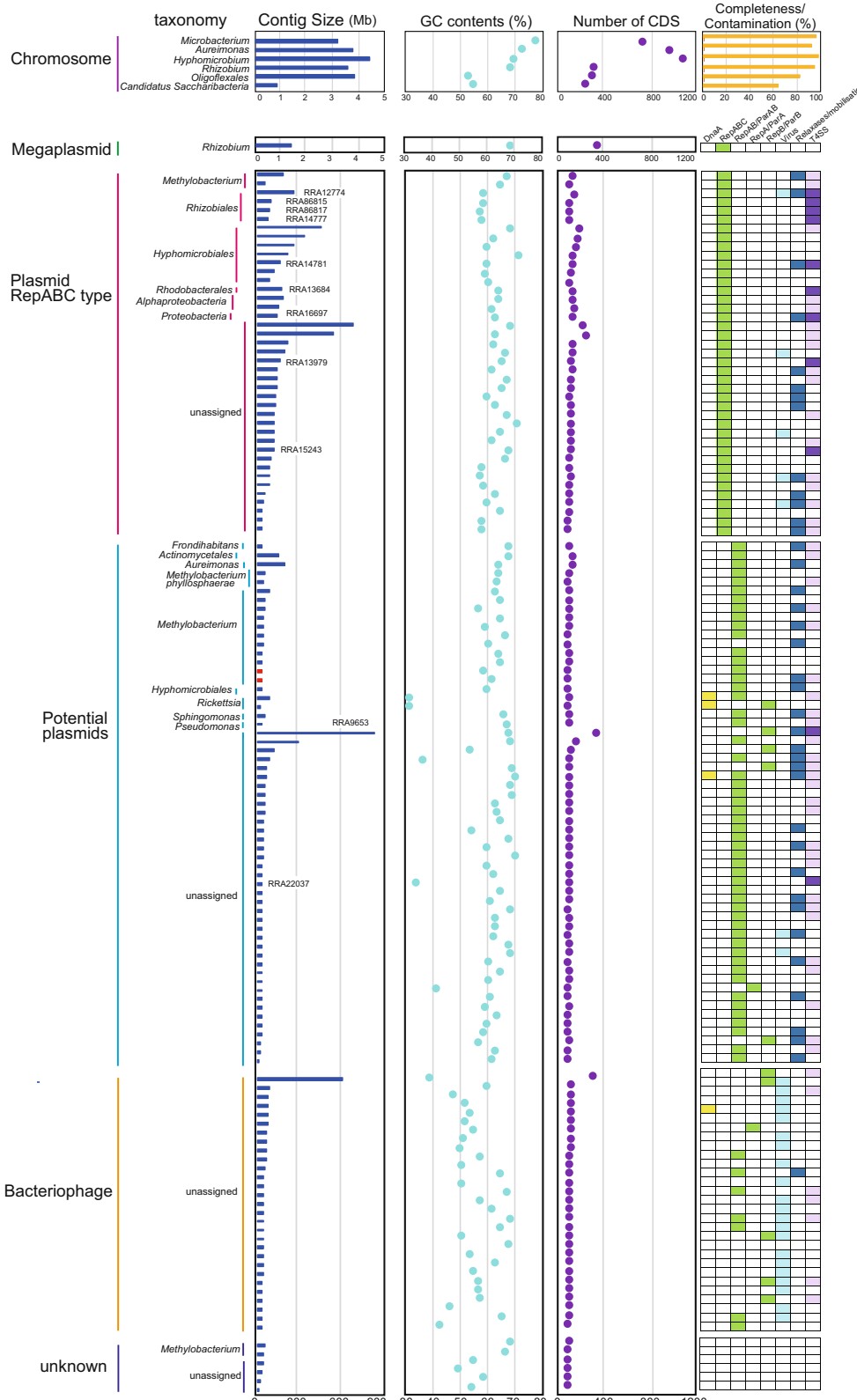

**Fig. 3 | Characteristics of circular contigs ($n$ = 142).** For chromosome, completeness (yellow) and contamination (orange) were shown. For megaplasmid, plasmid and bacteriophage, gene annotations are indicated by color blocks. Dark purple and light purple represent complete/nearly complete, and partial genes of VirB/VirD T4SS systems, respectively. Red bars in the contig size column indicate known plasmid sequences. The contig ID shown in the contig size column corresponds to contigs carrying complete/nearly complete VirB/VirD T4SS.

(Fig. 3). In particular, RRA944769 could be classified as a complete genome of a novel family of *Oligoflexales* based on 16S rRNA gene identity and ANI. Curiously, no 16S rRNA gene was detected in RRA2326, but it was assigned to the genus *Aureimonas* by ANI (Fig. 3). This confirms a previous report showing that *Aureimonas* sp. AU20, isolated from the rice phyllosphere, has its 16S rRNA gene on a small plasmid, but not on the chromosome[61]. Our data contribute to resolving ambiguities surrounding chromosomal rearrangements in such organisms during cultivation.

Of the 136 contigs under 1 Mbp, 134 presented sequences did not align with any known sequences, indicating its novelty. The two other contigs were aligned with high similarity to a plasmid in *Methylobacterium phyllosphaerae* strain CBMB27 (NZ_CP015369.1; the red color in contig size column in Fig. 3 and Supplementary Fig. 6). Among these smaller contigs, 61 genes across 41 contigs were annotated as *repC* (Fig. 3), suggesting that these contigs previously unidentified *repABC* plasmids. Further investigations into the plasmid hosts showed that sixteen of these contigs were associated with *Alphaproteobacteria*. However, the origins of the remaining 23 contigs could not be determined (Fig. 3 and Supplementary Fig. 7). These results suggest that nearly two-thirds of the *repABC* plasmids identified in this study is originated from bacteria species not previously known to carry *repABC* plasmids, revealing a significant aspect of microbial diversity and plasmid distribution.

In addition, 29 contigs were classified as dsDNA bacteriophages, each scoring highly (>0.8) using virsorter2, and exhibiting a range of CheckV completeness from 14.3 to 100% (Fig. 3). Of these, 21 contigs carried putative phage-related genes, such as those encoding for capsid proteins. Furthermore, 13 contigs contained genes for putative partitioning proteins, seven carried genes for VirB/VirD4 component, and one included a gene for the relaxosome protein TraY (Fig. 3). Despite identifying these potential new bacteriophages, their taxonomic classification remained elusive.

Additionally, we identified 59 contigs featuring genes commonly associated with VirB/VirD4 component, RepAB and relaxosome proteins. However, these contigs were not classified as either plasmids or bacteriophages by mob-typer and virsorter2. Given that genes for RepAB and relaxosome proteins, are typically found on plasmids rather than chromosomes[62,63], these contigs may represent plasmids (Fig. 3). Gene similarity searches on these contigs indicated that 21 out of the 59 were likely derived from *Alphaproteobacteria*, *Gammaproteobacteria* (specifically *Pseudomonas*), or *Actinobacteria* (Fig. 3).

Pathogenic and symbiotic bacteria utilize the VirB/VirD4 T4SS to manipulate host cell functions. In the case of *A. tumefaciens*, the 11 gene products of the *virB* operon, together with the VirD4 protein[64]. We found that genes for VirB/VirD4 T4SS components were present on 11 contigs (Fig. 4). Nine of these contigs were identified as *repABC* type plasmids, while the other two were also likely to be plasmids. The likely origins of these contigs were from *Proteobacteria*, specifically to the families *Rhizobiaceae*, and *Rhodobacterales* within the *Alphaproteobacteria* group. However, the remaining four contigs could not be assigned to any specific taxonomic group. A comparison of the gene arrangements on these 11 contigs with those of *A. tumefaciens* showed that all, or most components of the VirB/VirD4 T4SS were present, although the arrangement differed from that in *A. tumefaciens*, with some gene duplicated and others missing. Furthermore, an additional 52 contigs carried at least one component gene of the VirB/VirD4 T4SS (Fig. 3).

We identified five small circular contigs each carrying a presumptive *dnaA* gene, as typically a part of the DNA replication machinery in bacterial chromosomes (Fig. 3). A similarity analysis revealed that two of these contigs likely originated from *Rickettsia*, obligate intracellular α-proteobacteria known to associate with various eukaryotic hosts. Notably, approximately half of the 26 recognized *Rickettsia* species have plasmids, some harboring *dnaA*-like genes, varying in size from 12 to 83 kb[65]. We also detected *dnaA* genes on contigs that were classified as bacteriophage, potential plasmids, or from the *Candidatus Saccharibacteria* chromosome. Two other contigs traced back to *Methylobacterium* were identified, but it remained unclear whether these were plasmids or bacteriophages. Four other contigs could not be classified as chromosomes, plasmids, or bacteriophages due to the lack of recognizable similarity to known bacterial or bacteriophage-derived genes in public databases. Overall, our analysis tentatively identified one chromosome, 100 plasmids (41 *repABC*-type plasmids and 59 potentially plasmid-associated contigs), 29 bacteriophages, and 6 unclassified contigs (Fig. 3). This underscores the effectiveness of long-read metagenomic sequencing in identifying a substantial number of plasmids within a complex microbial community, most of which are novel.

## Taxonomic assignment of MAGs and large-size contigs

Except for the 142 circular contigs, we utilized 25,925 contigs in the binning procedure, resulting in 2205 contigs being grouped into 157 MAGs (Fig. 5). The genome sizes of these MAGs ranged from 274 kbp and 9.8 Mbp, with the GC contents varying from 36.4% to 75.1%, and all contigs were successfully assigned using ANI. At the class level, *Alphaproteobacteria*, *Planctomycetes* and *Actinobacteria* were predominant, representing 45.2%, 25.5%, and 8.9% of the MAGs, respectively. Notably, *Methylobacterium* was the most abundant genus, accounting for 17 of the MAGs, aligning with the bacterial community composition estimates derived from 16S rRNA gene analysis in the metagenome data. Furthermore, 38 MAGs showed >90% completeness and <5% contamination as assessed by CheckM, suggesting that these contigs likely represent nearly complete chromosomes including *Methylobacterium* (6 MAGs) and *Planctomycetes* (7 MAGs). Since *Planctomycetes* is still difficult to culture[66], our MAGs data could support for design the medium for isolation and cultivation of *Planctomycetes* based on the genomic data.

We also examined 37 large-size contigs (>1 Mbp) which were not included in our binning process (Supplementary Fig. 8). To determine the taxonomy of these large contigs, which ranged in size from 1 to 4.4 Mbp, we used a combination of 16S rRNA gene sequencing, ANI analysis via GTDB-tk and blast searches. Despite these efforts, 10 of these contigs could not be taxonomically assigned using these methods. However, the genes of 4 of these contigs showed high identity (≥95%) and length coverage (≥90%) to the genome of *Moesziomyces antarcticus*. Three of four contigs were carried minichromosome maintenance proteins (MCM2, 3, 6, and 10), suggesting that those three contigs may be a minichromosome of *M. antarcticus*. We also attempted to discriminate between chromosomal and plasmid contigs using ANI and the presence or absence of DNA replication initiators DnaA (for bacterial chromosomes). In total, 27 of the large contigs were classified as bacterial chromosome, 4 were fungal sequences, and the other 6 large contigs were not classified as either chromosomal, plasmids or yeast using these methods. Additionally, the 3 contigs were shown to have >90% completeness and <5% contamination by CheckM, suggesting that those contigs were nearly complete chromosomes.

## Complete genome of a bacterium in the *Candidatus Saccharibacteria* difficult-to-culture phylum

One of the key benefits of long-read metagenomic analysis is the potential to obtain complete genome sequences of uncultivable microorganisms. Here, we obtained the whole chromosome sequence of a member of the difficult-to-culture *Candidatus Saccharibacteria* phylum as a circular contig (RRA8490, Figs. 3 and 6). Phylogenetic analysis based on 16S rRNA genes indicated that RRA8490 clusters with isolates found in the human oral microflora (Supplementary Fig. 9A). A comparison of whole genome sequences and amino acid identities between RRA8490 and previously determined strains in the *Saccharibacteria*[67–72] showed that the genomes of these strains and RRA8490 were distinct, with average amino acid identities ranging from 52.2 to 54.2% (Supplementary Fig. 9B, C). Unlike other strains in *Candidatus Saccharibacteria* and *Candidatus Patescibacteria*, RRA8490 did not encode amino acid or fatty acids synthesis genes[72], but it did presumptively encode enzymes that metabolize glucose to ribose and glycerate-3-phosphate, as well as phosphoenol-pyruvate to malate, suggesting that these pathways may be used to generate ATP. RRA8490 also encoded four regions of type IV pili ($pilM_1N_1O_1B_1TC_1D$, $pilB_2C_2M_2N_2O_2$, $pilB_3$, $pilB_4$), similar to a previously reported *Saccharibacteria* (TM7) genome that carries

**Fig. 4 | Gene arrangements, predicted host, and estimated type of plasmid for T4SS genes discovered in the metagenome.** Purple indicates hypothetical or non-T4SS component genes.

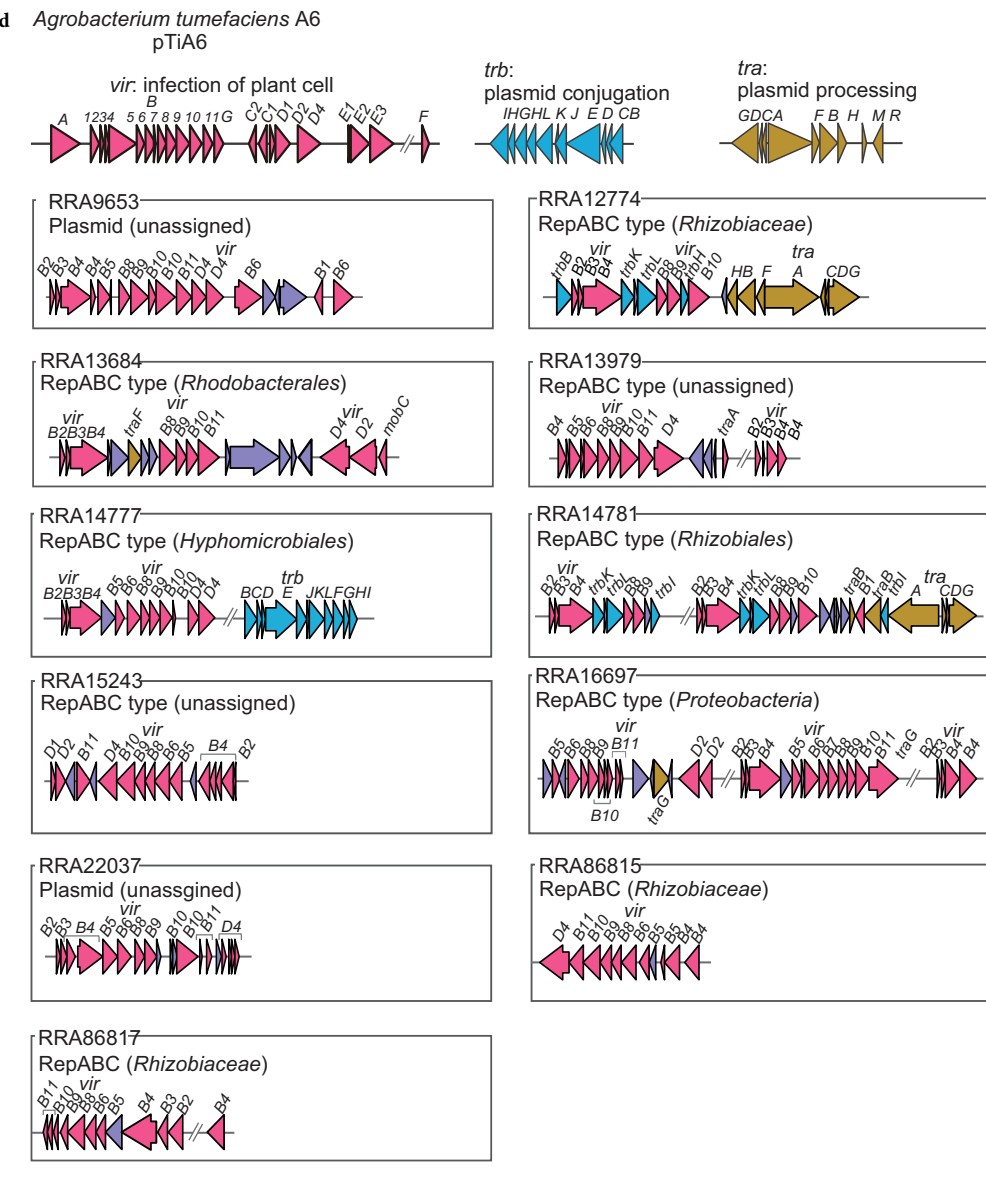

type IV pili for host cell attachment[68]. In addition to these characteristics, RRA8490 also encoded the cytochrome oxidase complex CyoABCDE, as previously reported (Fig. 6).

## Discussion

This study demonstrated that enzymatic genomic DNA extraction combined with long-read metagenomic sequencing is an effective tool for profiling plant microbiota and their genomes, as well as defining complete chromosomes, plasmids and bacteriophages from long, high quality contigs. Comparisons of the community-profiling datasets confirms that the enzymatic DNA extraction method is largely unbiased, with the notable exception of the inclusion of fungal DNA from the *Moesziomyces* genus. This is not surprising, as *Moesziomyces* spp. are commonly detected in plants[73,74], but not amplified by bacterial 16S primers. Our data indicate that *C. pusillum* is the dominant species in rice leaves (Supplementary Data), which is consistent with the fact that *Curtobacterium* spp. have been isolated from the leaves of many different plants[75–78] and are known to be abundant in a leaf litter communities[79]. Similarly, the *Methylobacteriaceae* family is a dominant presence (Fig. 1), as seen in the aerial parts of many plants[80]. The long-read metagenome provided detailed insights at the species level, enabling identification of the individual strains within these taxonomic ranks in the rice-phyllosphere. This hints at potential new roles for

microbiome members in rice-phyllosphere colonization. For instance, the genus *Methylobacterium*, comprising 45 recognized species[81], was represented by nine species in our long-read sequence data. While environmental factors such as plant growth stages might influence the presence and absence of certain species, long-read metagenomics offers a powerful means to estimate the functional capabilities of these organisms for adaptation in the rice phyllosphere.

The number and location of rRNA operons (*rrn*) can vary significantly among bacteria, and some bacteria have multiple copies of *rrn* on different chromosomes, as seen in *Brucella*[82] and *Vibrio*[83], as well as *E. acetylicum*. In contrast, long-read metagenomics-based identification of the precise number of 16S rRNA genes allows for accurate determination of bacteria abundance within a community. In addition, the relative abundance of 16S rRNA genes detected in the metagenome ranged from 0.02% to 11.2%, demonstrating the depth of coverage provided by long-read metagenomics. However, a comparison between bacterial species detected in the metagenome and nearly full-length 16S rRNA amplicon sequencing revealed that some species were only detected using one method or the other, despite having relative abundances greater than 0.02% (Supplementary Data). This highlights the importance of using a combination of long-read sequences, such as metagenomes, and 16S rRNA amplicon sequencing for more comprehensive taxonomic assignments. Overall, we accurately identified

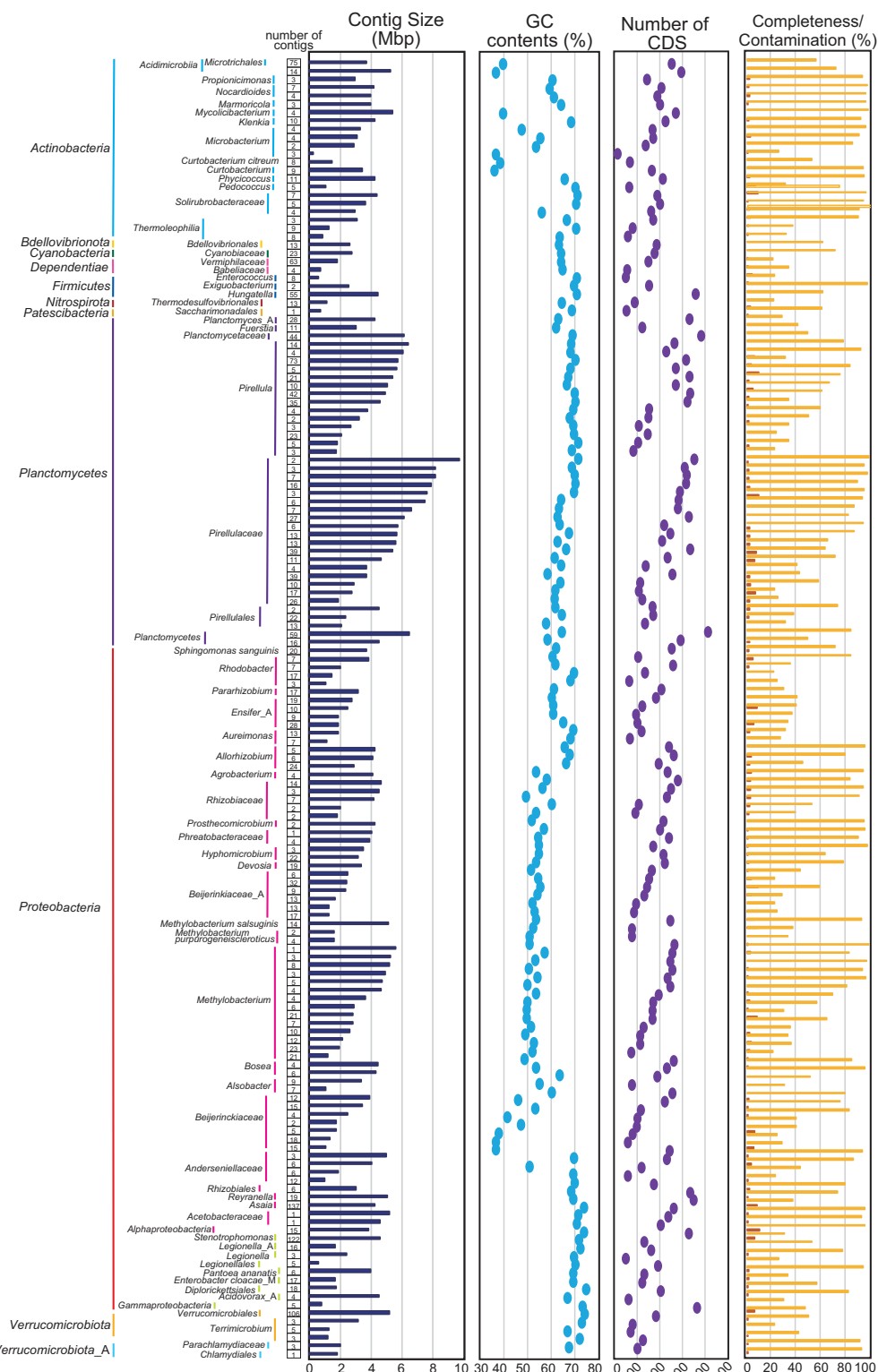

**Fig. 5 | MAGs constructed in this study.** The taxonomic assignment of each contig was determined by ANI. The completeness (yellow) and contamination (orange) were shown.

bacteria in the rice phyllosphere, with the potential for greater discrimination between organisms as new analysis methods become available.

A major advantage of long-read metagenomics over short-read metagenomics is that it can be used to define circular mobile elements such as plasmids and bacteriophages (Fig. 3). However, our understanding

of these elements has been limited. For example, the distribution of the VirB/VirD4 system is directly linked to plant host health and growth. T4SS are highly diverse[84–87], and the exact number of genes and their role in T4SS assembly or function is unknown in many classes of T4SS[84]. Our study showed that the five plasmids (RRA9653, RRA13979, RRA16697,

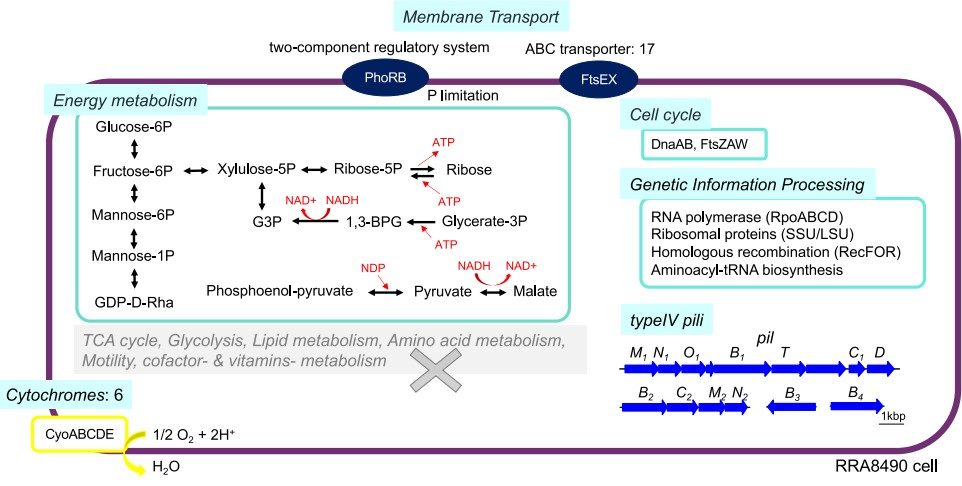

**Fig. 6 | Predicted metabolism of RRA8490, a potential new strain in the *Candidatus Saccharibacteria* phylum.** Metabolic pathways were reconstructed using kofamscan, Interproscan, and similarity searches.

RRA22037, and RRA86817) conserved VirB/VirD4 system essential for the assembly of this system in *A. tumefaciens*, suggesting that these five plasmids are the *A. tumefaciens*-type VirB/VirD4 system (Fig. 4). Interestingly, Ti plasmids belong to the *repABC* family, which is widely distributed among many species of *Alphaproteobacteria*. However, no *repABC* genes were found on two of the plasmids (RRA9653 and RRA22037 in Fig. 4), suggesting that these plasmids may have been horizontally transferred from other bacterial taxa. These results suggested that those might be widely distributed in the rice phyllosphere to modulate the host plant immunity to colonize the space. However, predicting the host of plasmids, particularly those with a broad host range, is a challenging task in metagenomic studies[13]. While we have proposed potential hosts for some plasmids, the origins of others remain uncertain. Future advancements, such as the use of droplet microfluidics for single bacterial cell isolation and plasmid-specific markers, could provide more comprehensive understanding of plasmid-host relationships[88].

Our analysis was able to define the complete circular genome (RRA8490) of difficult-to-culture bacterium belonging to the *Candidatus Saccharibacteria* phylum (Figs. 3 and 6). Members of this phylum have been detected in various natural environments such as soils, animals, and plants, but few cultured isolates have been limited our understanding of their biology[67–72]. Compared to the recently nearly completed genome (1.45 Mb) of an oat-associated member of the *Candidatus Saccharibacteria* phylum strain YM_S32, RRA8490 is much smaller (0.83 Mb) and belongs to a different clade (Supplementary Fig. 9). RRA8490 apparently lack the ability to synthesize amino acids from central metabolites, but RRA8490 is predicted to carry type IV pili and cytochrome bo3, similar to others in *Candidatus Saccharibacteria* phylum. RRA8490 is predicted to be able to assimilate and metabolize glucose and fructose, which are compounds found in leaf exudates of the rice phyllosphere[89], suggesting that RRA8490 may utilize these compounds as carbon sources. Additionally, *Candidatus Saccharibacteria* are obligate epibionts of *Actinobacteria*, which they lyse to obtain nutrients[68], suggesting that RRA8490 may not rely solely on plant exudates for nutrient acquisition, but may also degrade *Actinobacteria*. The CyoABCDE, cytochrome o oxidase complex, is used by *Rhizobium etli* to adapt to microaerobic conditions[90], but Cyo appears to be produced only under oxygen-rich growth conditions in *E. coli*[91]. These results suggest that the ability to function at a wide range of oxygen concentrations, as demonstrated by the presence of Cyo in RRA8490, would be beneficial for this bacterium as it adapts to a variety of oxygen conditions in its natural environment.

In conclusion, long-read metagenomics fueled by high-quality DNA extraction provides an efficient method for exploring uncharted organisms in the plant microbiome, and the resulting data represents an emerging primary resource for a deeper understanding of plant-associated microbial ecology.

## Data availability

The raw data and assembly data have been deposited in NCBI under accession number SAMN32580422 (BioSample), SRP420173 (SRA) and PRJNA929667 (BioProject).

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

## Acknowledgements

We are grateful to the technical staffs of the Department of Technical Development in ISAS of The University Tokyo for their maintenance of plant materials. We also thank Prof. Dr. K. Minamisawa (Tohoku University) and Dr. S. Hara (NARO) for sharing the cell-density centrifugation from rice plants protocol. This work was supported by the JSPS KAKENHI grants JP20H05909 and JP22H00364 (K.Sh.) and JP20H05592 (S.M.) and by JPNP18016 commissioned by the New Energy and Industrial Technology Development Organization (NEDO).

## Author contributions

S.M. and K.Sh. contributed to the design of the work. S.M., P.G., K.Sa., and K.Sh carried out rice sampling from an experimental field plot. S.M. and A.S. performed laboratory work. S.M., P.G., Y.K., M.A., K.Sa., W.I. and W.S., performed bioinformatics analysis. S.M. and K.Sh. wrote the manuscript. All authors interpreted the results and contributed to the final manuscript.

## Competing interests
The authors declare no competing interests.
