## [Peer Review File · Communications Biology]

This manuscript has been previously reviewed at another Nature Portfolio journal. This document only contains reviewer comments and rebuttal letters for versions considered at Communications Biology.

Reviewers' comments:

Reviewer #1 (Remarks to the Author):

This is interesting paper that obtaining data on different gDNA extraction methods and long read metagenome sequencing, augmented by full length amplicon sequencing (long read) and V4 region (short read) amplicon sequencing. The shotgun analysis covers both chromosomal sequence (including some CRP genomes) and non-chromosomal replicons. The results are valuable, both in their own right, and also because they clearly highlight the difficulties in calibrating community composition and structure across multiple sequencing modalities.

One obvious weakness with the analysis of the shotgun data is that it attempt the use of binning procedures to try and recover more genomes. As a number of previous papers have shown, while some genomes will drop out of an assembly algorithm as complete, even closed chromosomal sequence from long read metagenome, most will not, and it is clear that binning procedures will need to be adapted or designed for capturing still-fractionated genome sequence. For example, PacBio offers a suggested pipeline for recovery of long read MAGs:

<https://github.com/PacificBiosciences/pb-metagenomics-tools/blob/master/docs/Tutorial-HiFi-MAG-Pipeline.md>

Another example is the nanophase workflow:

<https://microbiomejournal.biomedcentral.com/articles/10.1186/s40168-022-01415-8>

for ONT data. Based on my own use of this workflow for single ONT datasets of ca. 90Gbp, I would predict it will greatly improve the rate of genome recovery from the current data set, and bring this analysis up to the edge of current practice.

Minor points

--p5, line 121. Aligned to SILVA db how?

--p6, around line 128. Is there any further details that need to be provided about the PacBio data processing?

--p6, line 132, “..Koshihikari [genome] is highly fragmented, with an average read length of 32 bp”: “read” should be “contig” and should “32 bp” be 32 kbp?

--Figure 3 and 4 are wonderful, but Fig.3 would improve will inclusion of a panel of CheckM results (completeness, contamination).

Reviewer #2 (Remarks to the Author):

General:

In this manuscript, Masuda and colleagues used long-read sequencing to study, in detail,

the rice leaf microbiome. This is a promising novel technology in metagenomics, as the resulting long-reads of typically 10-20Kb allow for a better recovery of (nearly) complete metagenome-assembled genomes (MAGs). To do that, the authors improved the enzymatic DNA extraction protocol in order to recover intact chromosomes. Metagenomic sequencing led to a total assembly size of 1.7 Gb, while they could recover 172 contigs larger than 1 Mb and 142 circular contigs, spanning complete chromosomes, plasmids and bacteriophages. This is the main novelty of this work, as from previous Illumina metagenomes, recovering such contigs was an almost impossible task. In addition, they also compared the 16S rRNA sequences recovered from three distinct approaches: from the assembly of the long-reads, from the sequencing of the full 16S rRNA gene amplicon using long-reads (PacBio Sequel II) and the widely used 16S rRNA V4 region short amplicon (Illumina MiSeq sequencing).

The main flaw of this manuscript in my view is that the authors have skipped a whole step in the recent history of metagenomics represented by short-read (Illumina) metagenomics instead they have abundantly used amplicon sequencing which is an older (some would say obsolete) technology. The manuscript is largely a comparison between amplicon sequencing and long-read metagenomics but both culture and short-read metagenomics have been nearly completely ignored. In addition, very little is concluded or explained about the biology of the leaf microbiome (rice or otherwise).

In the introduction, authors should describe what is known about taxonomic and genomic features of leaf microbiomes particularly (if any study is available) the one of rice. The first paragraph is too vague and largely predictable. What is expected to find on the leaves of plants? It is a strongly irradiated and dry (stressful) environment. There are many isolates from that habitat judging from the results obtained here and they should be described at least in summary form.

I also missed a proper introduction about the limitations of Illumina metagenomics that incidentally can also provide several fragments of 16S rRNA reads with no need for PCR (and its cognate bias).

I think that for this manuscript to be published it requires at least a different introduction and discussion to be written (the latter probably deleted). It also needs more Biology and less benchmarking of the long-read metagenomics that has already been extensively proven in the literature in several habitats and sets of microorganisms.

Specific comments:

Ln24 Considering the results I would say “identifying the bacterial species that colonize the plants by culture-independent approaches”

Ln 31 Specify low identity threshold between brackets

Ln 35 no need to repeat candidate twice “the phylum Candidatus Saccharibacteria would suffice

Ln 65 why is it limited?

Ln 72 “genome reconstruction” is not an objective of barcoding

Ln74 there is also the possibility of matching the 16S rRNA gene fragments recovered from short reads to 16S rRNA databases to provide a taxonomic profile not based on PCR products (many examples in the literature).

Line 75 onwards short-read metagenomics once assembled can reconstitute plasmids and nearly complete genomes. It fails in reconstructing the flexible genome (that is, regions that vary from one

strain to another and are simultaneously present in the sample) due to assembly issues.

Line Ln89 onwards. It is not clear to me why the authors went to the trouble to do a tag sequencing of 16S rRNA genes instead of an Illumina run. In any case, it would have been interesting to compare the taxonomic profile provided by the three methods from the 16R rRNA fragments provided by each with the one of long reads that generates complete 16S rRNA genes without any assembly required. Assembly would likely improve with an Illumina metagenome to correct the error rate of PacBio.

Ln 157 why >98.7% how is this threshold justified? Typically for species, it is 97%

Ln167 again 98.7%

Ln 173 please explain how ANI was calculated

Ln 230 onwards this section might be better as a short paragraph of the methods section and data not shown

Ln246 gDNA?

Ln 250 I wonder what happened with the remaining 11,878 contigs? I suggest the authors perform metagenomic binning to group the smaller contigs into larger units (eventually even MAGs) and also to try to improve the 168 large >1 Mb) contigs.

Ln 251 Again, the availability of an Illumina metagenome would have been very helpful because using coverage by the same reads used by assembly will strongly bias towards easy-to-assemble (e.g. with low intraspecies diversity) genomes. Thus, it is not surprising that “all large size contigs (≥ 1 Mbp) were of high quality”.

Ln 271 I was again puzzled by the 98.7% threshold. Where does it come from?

Ln 279 whole paragraph. The species names by themselves do not mean much to a non-specialist please provide a brief description of each, e.g. the known plant growth promoter *Methylobacterium*, and so on.

Ln290 whole paragraph. I don't see the point of doing again whole 16S amplicons (an inherently more biased method due to primers and PCR) to check the community structure. But it would suffice to say that this method provided a similar taxonomic profile. The same applies to the next paragraph (Ln 301). That long-read metagenomes provide more reliable community structure than amplicon sequencing has been abundantly proven in published papers already.

Ln 316 the annotation rate sounds very poor (less than 15%), they have only used COGs, maybe using Pfam, Kegg or other databases might provide a more satisfying output.

Ln 325 I miss some comparison to previous studies. Was the abundance of *Methylobacterium* unexpected? Again, here the lack of a proper introduction is patent again.

Ln 338. It would be nice to know something about the remaining 15 contigs that are not yeast.

Ln354 whole paragraph. With high ANI the authors should focus their attention on the coverage i.e. percentage of orthologous genes between genomes. It would also be nice to have some whole genome alignment.

Ln 394 whole paragraph. Both VirB and toxin-antitoxin systems are widespread in chromosomes of many bacteria, certainly, their presence is not enough to classify a contig as a plasmid.

Ln 443 *Saccharibacteria* have been obtained in monoclonal cultures so I would avoid the term “as-yet-uncultured”

Ln 445 peptidoglycan is present in nearly all bacteria regardless of the Gram stain

Ln448 inability to synthesize fatty acids and amino acids is widespread in *Patescibacteria*

Discussion. Whole section. Please avoid repeating results in this section. As written it is largely a repetition of results or presenting results not stated before.

Figure 3. Representing the read depth using logarithmic values can help to visualize the low abundant microbes.

Figure 3. The number of CDS predicted for the four fungi contigs are too small. Is it due to a bias in the gene prediction using prokka? This should be indicated elsewhere.

Figure 4. The dots indicating the number of CDS do not correspond to the contig size. It seems that the data is displaced.

Reviewer #3 (Remarks to the Author):

The manuscript of Masuda et al reports on a study which aimed to uncover plant microbiome by using long-read metagenomic sequencing. The manuscript is very well written, figures are informative and high quality. The information content of the manuscript is also high and its novelty is obvious. In my opinion, the manuscript can be accepted as it is, no modifications are required.

Reviewers' comments:

Reviewer #1 (Remarks to the Author):

This is interesting paper that obtaining data on different gDNA extraction methods and long read metagenome sequencing, augmented by full length amplicon sequencing (long read) and V4 region (short read) amplicon sequencing. The shotgun analysis covers both chromosomal sequence (including some CRP genomes) and non-chromosomal replicons. The results are valuable, both in their own right, and also because they clearly highlight the difficulties in calibrating community composition and structure across multiple sequencing modalities.

One obvious weakness with the analysis of the shotgun data is that it attempt the use of binning procedures to try and recover more genomes. As a number of previous papers have shown, while some genomes will drop out of an assembly algorithm as complete, even closed chromosomal sequence from long read metagenome, most will not, and it is clear that binning procedures will need to be adapted or designed for capturing still-fractionated genome sequence. For example, PacBio offers a suggested pipeline for recovery of long read MAGs:

<https://github.com/PacificBiosciences/pb-metagenomics-tools/blob/master/docs/Tutorial-HiFi-MAG-Pipeline.md>

Another example is the nanophase workflow:

<https://microbiomejournal.biomedcentral.com/articles/10.1186/s40168-022-01415-8>

for ONT data. Based on my own use of this workflow for single ONT datasets of ca. 90Gbp, I would predict it will greatly improve the rate of genome recovery from the current data set, and bring this analysis up to the edge of current practice.

Thank you very much for the comment. We reconstructed the MAGs according to a previous paper (Kato et al., 2022, <https://doi.org/10.3389/fmicb.2022.1045931>). In the binning procedure, many of the >1Mb contigs (129/172 contigs) were binned. Thus, removed the previous Fig.3, and created the new figure (Fig. 5) for MAGs data. We removed the Supplementary Table 4 which summarized the >1Mbp contigs. We created

the new figure (Supplementary Fig. 8) for the >1Mbp contigs (37 contigs) which were not binned. According to above, we combined the six circular contigs (>1Mbp) shown in previous Fig. 3 to Fig. 4, and created the new Fig. 3 as circular contigs in this study.

Minor points

--p5, line 121. Aligned to SILVA db how?

We used Qiime2 for truncation, alignment and phylogenetic annotation of the sequences.

We changed the sentences (L131-134)

--p6, around line 128. Is there any further details that need to be provided about the PacBio data processing?

No, we just removed the contaminant sequences such as plant genome and PacBio's internal control sequences as described parameters in this manuscript.

--p6, line 132, “.Koshihikari [genome] is highly fragmented, with an average read length of 32 bp”: “read” should be “contig” and should “32 bp” be 32 kbp?

Thank you very much for the comment. The Koshihikari genome was sequenced using Solexa, and the reads were mapped to the Nipponbare genome. Finally, the authors generated the Koshihikari consensus genome with length ranging from 32 to 40,797 bp, and averaging 468 bp. We changed the average read length from 32 bp to 468 bp and cited the paper (L145)

--Figure 3 and 4 are wonderful, but Fig.3 would improve will inclusion of a panel of CheckM results (completeness, contamination).

Thank you very much for the comment. According to the creation of new figure, we included the CheckM result in Fig.3 for complete chromosome, Fig.5 for MAGs and Fig. S8 for >1Mbp contigs.

Reviewer #2 (Remarks to the Author):

General:

In this manuscript, Masuda and colleagues used long-read sequencing to study, in detail, the rice leaf microbiome. This is a promising novel technology in metagenomics, as the resulting long-reads of typically 10-20Kb allow for a better recovery of (nearly) complete metagenome-assembled genomes (MAGs). To do that, the authors improved the enzymatic DNA extraction protocol in order to recover intact chromosomes. Metagenomic sequencing led to a total assembly size of 1.7 Gb, while they could recover 172 contigs

larger than 1 Mb and 142 circular contigs, spanning complete chromosomes, plasmids and bacteriophages. This is the main novelty of this work, as from previous Illumina metagenomes, recovering such contigs was an almost impossible task. In addition, they also compared the 16S rRNA sequences recovered from three distinct approaches: from the assembly of the long-reads, from the sequencing of the full 16S rRNA gene amplicon using long-reads (PacBio Sequel II) and the widely used 16S rRNA V4 region short amplicon (Illumina MiSeq sequencing).

The main flaw of this manuscript in my view is that the authors have skipped a whole step in the recent history of metagenomics represented by short-read (Illumina) metagenomics instead they have abundantly used amplicon sequencing which is an older (some would say obsolete) technology. The manuscript is largely a comparison between amplicon sequencing and long-read metagenomics but both culture and short-read metagenomics have been nearly completely ignored. In addition, very little is concluded or explained about the biology of the leaf microbiome (rice or otherwise).

In the introduction, authors should describe what is known about taxonomic and genomic features of leaf microbiomes particularly (if any study is available) the one of rice. The first paragraph is too vague and largely predictable. What is expected to find on the leaves of plants? It is a strongly irradiated and dry (stressful) environment. There are many isolates from that habitat judging from the results obtained here and they should be described at least in summary form.

I also missed a proper introduction about the limitations of Illumina metagenomics that incidentally can also provide several fragments of 16S rRNA reads with no need for PCR (and its cognate bias).

I think that for this manuscript to be published it requires at least a different introduction and discussion to be written (the latter probably deleted). It also needs more Biology and less benchmarking of the long-read metagenomics that has already been extensively proven in the literature in several habitats and sets of microorganisms.

Thank you very much for the comments. We changed the whole paragraph of introduction according to your comments. In introduction section, we explained the importance of phyllosphere-microbiome to the host plant, and the data obtained from culture-dependent and short-read approaches.

Specific comments:

Ln24 Considering the results I would say “identifying the bacterial species that colonize the plants by culture-independent approaches”

Thank you very much for the comment. As the comment, culture-dependent method could answer those questions, but culture-dependent method could not isolate all of the colonized bacteria in the plant. Thus, we changed the sentence that includes general questions (L23).

Ln 31 Specify low identity threshold between brackets

Thank you very much for the comment. I added (97%) to the sentence (L31).

Ln 35 no need to repeat candidate twice “the phylum Candidatus Saccharibacteria would suffice

Thank you very much for the comment. I considered your comment as you mentioned below and changed the sentence (L35) from “uncultured-bacteria” to “a difficult-to-culture bacterium”.

Ln 65 why is it limited?

Because of the complexity of plasmid sequence and bioinformatics tools for next generation sequencing. Although thousands of plasmids have been sequenced and assembled from not only isolated bacteria but also metagenomic data, constructing complete plasmid sequences from short reads remains a hard challenge. Plasmids represent a small fraction of the sample’s DNA and thus may not be fully covered by the read data in high-throughput sequencing experiments. Also, they often share sequences with the bacterial genomes and with other plasmids, resulting in tangled assembly graphs. In addition, because of the presence of numerous repetitive elements, identifying circulating plasmids are difficult to accurately assemble their whole genome sequences. For these reasons, plasmid assembly are usually incomplete, fragmented into multiple contigs, and contaminated with sequences from other sources (Pellow et al., 2021, Microbiome 9:144). We agree with those opinion and explained in L83-93.

Ln 72 “genome reconstruction” is not an objective of barcoding

Thank you very much for the comment. We removed the words “genome reconstruction” and explained those in the sentence (L62-67).

Ln74 there is also the possibility of matching the 16S rRNA gene fragments recovered from short reads to 16S rRNA databases to provide a taxonomic profile not based on PCR products (many examples in the literature).

Thank you very much for the comment. We explained those in the sentences (L65-67).

Line 75 onwards short-read metagenomics once assembled can reconstitute plasmids and nearly complete genomes. It fails in reconstructing the flexible genome (that is, regions that vary from one strain to another and are simultaneously present in the sample) due to assembly issues.

Thank you very much for the comment. We explained those in the sentences (L65-67).

Line Ln89 onwards. It is not clear to me why the authors went to the trouble to do a tag sequencing of 16S rRNA genes instead of an Illumina run. In any case, it would have been interesting to compare the taxonomic profile provided by the three methods from the 16S rRNA fragments provided by each with the one of long reads that generates complete 16S rRNA genes without any assembly required. Assembly would likely improve with an Illumina metagenome to correct the error rate of PacBio.

Thank you very much for the comment. Full-length of 16S rRNA gene amplicon sequencing was performed because we would like to compare the results among those methods (long-read metagenome, Full-length of 16S rRNA gene amplicon sequencing and short-read) as you mentioned. We described it in L302-304.

Ln 157 why >98.7% how is this threshold justified? Typically for species, it is 97%

Thank you very much for the comment. We cited the paper which 98.7% is the threshold as species.

Thank you very much for the comment. We changed the threshold from 98.7% to 97%, and explained "widely used for bacterial species definition" in L280. When we changed the threshold to 97%, the number of species were changed in the section of Estimation of microbial composition using 16S rRNA genes in long-read metagenomics in results section.

Ln167 again 98.7%

According to the above comment, we changed the threshold from 98.7% to 97%.

Ln 173 please explain how ANI was calculated

GTDBtk also calculated ANI value.

Ln 230 onwards this section might be better as a short paragraph of the methods section and data not shown

Thank you very much for the comment. We combined some sentence into the Materials and Methods section and the shortened the paragraph (12 lines to 9 lines).

Ln246 gDNA?

We changed the word from gDNA to genomic DNA (L254).

Ln 250 I wonder what happened with the remaining 11,878 contigs? I suggest the authors perform metagenomic binning to group the smaller contigs into larger units (eventually even MAGs) and also to try to improve the 168 large >1 Mb) contigs.

Thank you very much for the comment. We constructed the MAGs as described previously (Kato et al., 2022, <https://doi.org/10.3389/fmicb.2022.1045931>). Totally, 2,273 contigs were binned into 157 MAGs. The 157 MAGs were contained 129 of the 172 >1Mb contigs. Thus, we removed the previous Fig.3, and created the new figure (Fig. 5) for MAGs data.

According to construction of the MAGs, we combined circular contigs (>1Mb) in previous Fig. 3 into previous Fig. 4 and created new Fig. 3 for circular contigs. We also created the new figure (Fig. S8) which the >1Mbp contigs (37 contigs) were not binned. We removed the supplementary table 4 which was summarized >1Mbp contigs.

Ln 251 Again, the availability of an Illumina metagenome would have been very helpful because using coverage by the same reads used by assembly will strongly bias towards easy-to-assemble (e.g. with low intraspecies diversity) genomes. Thus, it is not surprising that “all large size contigs (\geq 1Mbp) were of high quality”.

We did not know the accuracy of the contigs constructed by PacBio long-reads in plant-microbiome because PacBio long reads showed the more errors than short-reads at this time (L259-262). So, we calculated the depth of the contigs according to a previous report which showed the contigs more than 5 depth as high accurate contigs constructed by PacBio. We showed that most of the reads were more than 5 depth and many of them were used to construct the large size contigs. Thus we would like to show that our data was reliable for the whole analysis in this study.

Ln 271 I was again puzzled by the 98.7% threshold. Where does it come from?

According to the above comment, we changed the threshold from 98.7% to 97%.

Ln 279 whole paragraph. The species names by themselves do not mean much to a non-specialist please provide a brief description of each, e.g. the known plant growth promoter *Methylobacterium*, and so on.

Thank you very much for the comment. We added the description of each bacterium in the sentence when the bacterium is described in the first time in the manuscript.

Ln290 whole paragraph. I don't see the point of doing again whole 16S amplicons (an inherently more biased method due to primers and PCR) to check the community structure. But it would suffice to say that this method provided a similar taxonomic profile. The same applies to the next paragraph (Ln 301). That long-read metagenomes provide more reliable community structure than amplicon sequencing has been abundantly proven in published papers already.

Thank you very much for the comment. We added the sentence “We compare the taxonomic profile of bacterial community composition based on the long-read metagenome and amplicon sequencing including full-length 16S rRNA gene using universal primers and the v4 region of short-read 16S rRNA sequencing.” In L302-305.

Ln 316 the annotation rate sounds very poor (less than 15%), they have only used COGs, maybe using Pfam, Kegg or other databases might provide a more satisfying output.

We annotated the genes using Pfam and Kegg, and compared the annotated genes among three databases (COG, Pfam and Kegg). The Venn diagram showed that almost all genes could be annotated using COG database. The genes just annotated using Pfam and Kofamscan were at 6% and 0.218% of total genes, respectively. Thus we considered that the genes annotated using COG database was provided us the gene information of rice-phylosphere metagenome.

Ln 325 I miss some comparison to previous studies. Was the abundance of Methylobacterium unexpected? Again, here the lack of a proper introduction is patent again.

Thank you very much for the comment. We explained that Methylobacterium is known as the dominant species in plant-phylosphere (including rice phyllosphere) in Introduction section (L59-60).

Ln 338. It would be nice to know something about the remaining 15 contigs that are not yeast.

Based on the blast search, we could not conclude taxonomy of those contigs. Thus, we explained in the L442 “the other 6 large contigs were not classified as either chromosomal, plasmids or yeast using these methods.”.

Ln354 whole paragraph. With high ANI the authors should focus their attention on the coverage i.e. percentage of orthologous genes between genomes. It would also be nice to have some whole genome alignment.

Thank you very much for the comment. We used both BLAST analysis and ANI to assign the taxonomy of the contigs. We explained in the sentences L430.

Ln 394 whole paragraph. Both VirB and toxin-antitoxin systems are widespread in chromosomes of many bacteria, certainly, their presence is not enough to classify a contig as a plasmid.

Thank you very much for the comment. We removed the “toxin-antitoxin systems” and “VirB/VirD4 component” from Fig. 3 and from corresponding sentence. In addition, carrying RepAB could be the evidence of the contig as plasmid because those contigs are circularized. Thus, we changed the sentence from “These genes are more commonly plasmid-borne than chromosomal” to “These genes, RepAB and relaxosome protein, are more likely plasmid-borne than chromosomal”.

Ln 443 Saccharibacteria have been obtained in monoclonal cultures so I would avoid the term “as-yet-uncultured”

Thank you very much for the comment. We replaced the word “as-yet-uncultured” to “difficult-to-culture” (L434). We also replaced and removed the uncultured/uncultivable from the sentences (L434-439).

Ln 445 peptidoglycan is present in nearly all bacteria regardless of the Gram stain

Thank you very much for the comment. We removed the corresponded sentence and illustration from Fig. 6.

Ln448 inability to synthesize fatty acids and amino acids is widespread in Patescibacteria

Thank you very much for the comment. We added the sentence “Unlike other strains in Candidatus Saccharibacteria and Candidatus Patescibacteria,” in L458-459

Discussion. Whole section. Please avoid repeating results in this section. As written it is largely a repetition of results or presenting results not stated before.

Thank you very much for the comment. We removed the repeated sentence described in results section from Discussion section.

Figure 3. Representing the read depth using logarithmic values can help to visualize the low abundant microbes.

Thank you very much for the comment. We changed the read depth as logarithmic value in Fig. S8.

Figure 3. The number of CDS predicted for the four fungi contigs are too small. Is it due to a bias in the gene prediction using prokka? This should be indicated elsewhere.

We predicted the genes of four fungi contigs using Augustus (L147-148).

Figure 4. The dots indicating the number of CDS do not correspond to the contig size. It seems that the data is displaced.

Thank you very much for the comment. We carefully checked and illustrated in the Figures 3, 4 and Fig. S8.

Reviewer #3 (Remarks to the Author):

The manuscript of Masuda et al reports on a study which aimed to uncover plant microbiome by using long-read metagenomic sequencing. The manuscript is very well written, figures are informative and high quality. The information content of the manuscript is also high and its novelty is obvious. In my opinion, the manuscript can be accepted as it is, no modifications are required.

Thank you very much for the comment.

REVIEWERS' COMMENTS:

Reviewer #1 (Remarks to the Authors):

The authors have addressed my concerns and I feel the paper has substantially improved in terms scientific content and clarity of presentation.

Reviewer #2 (Remarks to the Author):

The manuscript can be now accepted. However, the authors should take care of some small mistakes such as using the term *Candidatus* in italics (right) and the following name without (often wrong in the manuscript)

Reviewer #3 (Remarks to the Author):

The authors did a good job during the review of their manuscript.